# Ferromagnetic particles as magnetic resonance imaging temperature sensors

J.H. Hankiewicz[1], Z. Celinski[1], K.F. Stupic[2], N.R. Anderson[1] & R.E. Camley[1]

Magnetic resonance imaging is an important technique for identifying different types of tissues in a body or spatial information about composite materials. Because temperature is a fundamental parameter reflecting the biological status of the body and individual tissues, it would be helpful to have temperature maps superimposed on spatial maps. Here we show that small ferromagnetic particles with a strong temperature-dependent magnetization, can be used to produce temperature-dependent images in magnetic resonance imaging with an accuracy of about 1 °C. This technique, when further developed, could be used to identify inflammation or tumours, or to obtain spatial maps of temperature in various medical interventional procedures such as hyperthermia and thermal ablation. This method could also be used to determine temperature profiles inside nonmetallic composite materials.

[1] UCCS Center for the BioFrontiers Institute, University of Colorado, Colorado Springs, Colorado 80918, USA. [2] National Institute of Standards and Technology, Boulder, Colorado 80305, USA. Correspondence and requests for materials should be addressed to J.H.H. (email: jhankiew@uccs.edu).

Magnetic resonance imaging (MRI) is most commonly associated with images in living creatures where it can provide detailed information on the proton distribution, thus reflecting the nature of different tissues. However, it has multiple non-biological uses. For example, it has been used to measure the properties of solid rocket propellants and industrial polymers[1], the distribution of water in cements, ceramics and rocks[2] and the diffusion of $CO_2$ in porous media[3]. Here we report a method we developed to make temperature maps within an object using MRI. This could give, for example, the temperature profile in a laser heated material or the temperature profile in actively cooled electronics.

Of course, temperature is also important in living systems. Temperature is a fundamental parameter reflecting the biological status of the body and individual tissues. Clinical studies indicate that localized temperature measurements could be a useful method for the detection of a variety of health problems including certain tumours and inflammation[4,5].

The precise determination of tissue temperature is also important in various thermal medical interventional procedures. In hyperthermia therapy for selective tumour treatment, the temperature of tumour-affected tissue is raised for a prolonged time to 40–43 °C to induce apoptosis[6]. Thermal ablation procedures such as laser, radio-frequency, microwave and high-intensity focused ultrasound therapies utilize much higher temperatures (48–100 °C) for tissue necrosis through thermal coagulation[7]. The exact value of the applied temperature depends on the type of disease, heating modality, target size and position, and tissue heat conduction and absorption[8]. Monitoring temperature during standard magnetic resonance imaging is also critical because tissue around medical metallic implants can be overheated by eddy currents generated in the implant by fast switching magnetic gradients and radio-frequency pulses[9–12]. Outside of clinical applications, three-dimensional temperature measurements have been used in investigation of various food thermal processes[13,14].

Conventional thermometry is usually invasive, allows only single point temperature measurements, and may interfere with the therapeutic and imaging instruments. The ability to do *in vivo* monitoring of temperature in three dimensions is thus important for both diagnosis and treatment of patients. These limitations could be addressed using a minimally invasive magnetic resonance thermometry that produces high thermal, spatial and temporal resolution temperature maps superimposed on anatomical images within the targeted tissue.

There have been earlier schemes used to measure temperature in MRI. Some methods are based on changes in physical parameters with temperature. These include changes in the proton resonance frequency (PRF), diffusion coefficients, or $T_1$ and $T_2$ nuclear relaxation time[15,16]. PRF accuracy is about 1 °C in immobile tissue[17,18]. However, this method relies on a comparison to a baseline image, and is sensitive to motion[19] and thermally induced susceptibility artifacts during scanning[20]. This prevents highly accurate *in vivo* MRI thermometry[21,22].

MR thermometry with temperature-sensitive contrast agents has also been attempted[23–26]. The thermosensitive liposomes materials can measure absolute temperature[27], however, these agents basically 'light up' at a single temperature and do not give meaningful results over a range of temperatures[28].

We suggest a different mechanism may provide temperature information within MRI. If one adds ferromagnetic particles with a magnetization that is strongly temperature dependent, one can obtain a temperature-dependent linewidth in nuclear magnetic resonance (NMR) and consequent changes in MRI intensities, particularly in $T_2^*$ measurements[29]. Other studies have also employed ferromagnetic particles in MRI. Two involved small magnetic structures which create a local change in the resonance frequency[30,31]. Additionally, ferromagnetic nanoparticles have been used as contrast agents[32]. However, none of these earlier studies addressed temperature issues.

In the following paper, we demonstrate the creation of a new method of temperature measurement within tissues or other composite materials. The key idea is to use a ferromagnetic material where the magnetization changes rapidly near the temperature range of interest. The ferromagnetic particles embedded in the tissue will create a local dipole magnetic field that makes the static magnetic field of the MRI scanner inhomogeneous and, as a result, broadens the NMR line. This broadening will be temperature-dependent if the magnetic particles exhibit a rapid change of magnetization as a function of temperature. One way to accomplish this is to use a ferromagnetic material near its Curie temperature, $T_C$. Using gadolinium particles in an agar gel as an example, we can measure temperature changes on the order of one degree due to the induced brightness changes in the $T_2^*$ weighted MRI images.

## Results

**Characterization of Gd particles.** In our experiments, we employed gadolinium particles prepared by a mechanical method. Using scanning electron microscope (SEM) images, we estimated the particle's major axis average length as 4.8 μm with a standard deviation of 2.7 μm. An SEM image of the particles and the corresponding histogram with the major axis length distribution is shown in Fig. 1.

To determine the stability of the Gd particles in an aqueous solution, the NMR linewidth (measurements discussed below) of Ringer's solution-agar gel, with a 2.75 mM l$^{-1}$ concentration of Gd, was monitored over 20 months. A gradual decrease of the linewidth, by 8 Hz per month, was determined.

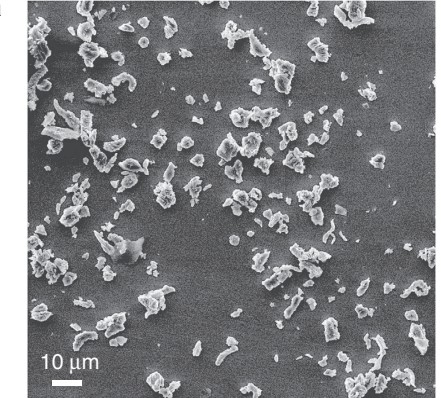

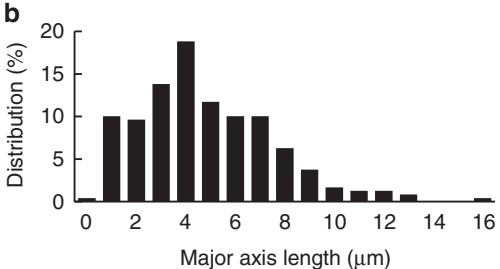

**Figure 1 | Properties of Gd particles.** (**a**) Scanning electron microscope image of particles. (**b**) Histogram of particles' major axis length distribution as calculated from a scanning electron microscope image.

**Magnetization measurements.** In zero applied field, Gd is characterized by a transition from the ferromagnetic to a paramagnetic state around 20 °C (ref. 33). As we will see, in the large fields produced in typical MRI systems (1.5 and 3.0 T), the transition is shifted to higher temperatures and broadened out. Gd also possesses a large magnetic moment, creating large changes in the NMR linewidth. The use of Gd for this application was suggested earlier in a study using a solid Gd wire[34]. Our theoretical calculations for magnetization as function of temperature and field, $M(T,H)$, are in good agreement with the experiments and show that the dipolar field produced by the ferromagnetic particles increases the NMR linewidth as temperature is reduced.

Figure 2 shows the results of magnetization measurements of metallic gadolinium particles using a superconducting quantum interference device (SQUID) at selected magnetic fields and the corresponding theoretical calculations of magnetization. The temperature-dependent measurements of the magnetization were carried out at fields of 364 mT, used later in the NMR measurements, and at fields of 1.5 and 3.0 T, typical in clinical MRI settings. The Curie temperature of Gd particles was determined to be 19 °C using the magnetization measurements at 1 mT. It is clear from Fig. 2 that an increase of magnetic field shifts the transition to the paramagnetic state towards higher temperatures and potentially makes gadolinium particles a useful temperature-sensitive contrast within our projected target range.

The temperature dependence of the magnetization may be calculated by a simple theoretical model. Within a mean field theory, the thermal averaged magnitude of a spin, $S$, is given by:

$$\langle S \rangle = S B_s(x) \qquad (1)$$

where $B_s$ is the Brillouin function; $x$ is the ratio of the magnetic energy to the thermal energy given by:

$$x = \frac{g\mu_B S(H + \lambda \langle S \rangle)}{kT}. \qquad (2)$$

Here $H$ is the applied field; $g$ is the Landé factor; $\mu_B$ is the Bohr magneton; $k$ is Boltzmann's constant; $T$ is temperature; and $\lambda \langle S \rangle$ measures the exchange field produced on a given spin. The exchange constant $\lambda$ is found from the experimental Curie temperature. equations (1) and (2) are solved self-consistently. The results, presented in Fig. 2, are in good agreement with the experimental measurements, indicating an understanding of the

origin and behaviour of the magnetization as a function of temperature and applied field. There is a single fitting parameter for all the theoretical curves, a reduction factor of 5.1 for the Gd magnetization from its bulk value. This will be discussed in more detail later.

**NMR linewidth broadening.** NMR measurements were carried out on Ringer's solution-agar gels with different concentrations of Gd in an applied field of 364 mT. We show both experimental and theoretical temperature-dependent linewidth broadening in Fig. 3.

The maximum Gd concentration in Ringer's solution-agar gel was 2.75 mM1[−1]. The results of the relative line broadening were obtained by subtracting the NMR linewidth at full-width at half maximum (FWHM) in pure agar gel from the FWHM linewidth in agar gel with suspended Gd particles. There are two key conclusions: (1) the addition of Gd produces a measureable, temperature-dependent change in the linewidth, and (2) significantly different concentrations all produce a temperature-dependent linewidth, indicating that the method is robust in terms of its sensitivity to concentration.

We can test our understanding of the origin of the temperature dependence of the NMR linewidth by a simple theoretical model. In general, linewidth calculations can be quite complex involving

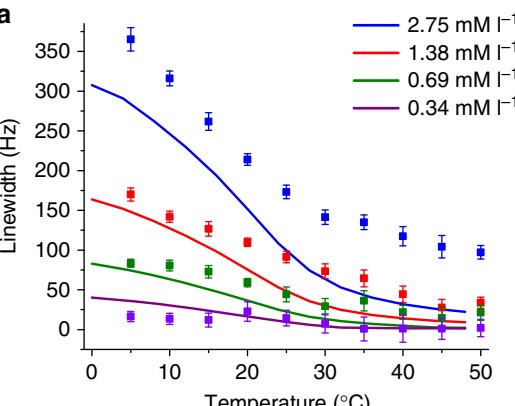

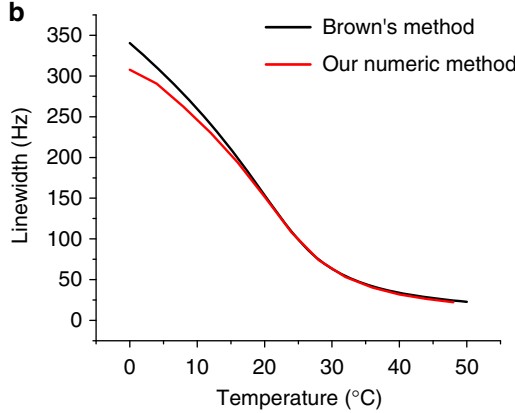

**Figure 3 | Temperature dependence of nuclear magnetic resonance linewidth broadening. (a)** Mean values and standard deviations of the experimental linewidth broadening in a mixture of 1% Ringer's solution-agar gel with four different concentrations of suspended Gd particles. Squares—data obtained from averaging of Fourier transforms of five nuclear magnetic resonance measurements. Solid lines—corresponding theoretical calculations. **(b)** Comparison of theoretical calculations with Brown's method[39] for the highest Gd concentration. The applied magnetic field is 364 mT.

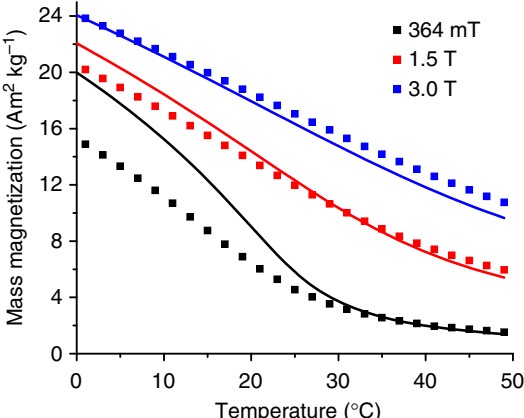

**Figure 2 | Magnetization of Gd particles versus temperature.** Data from superconducting quantum interference device measurements (squares) and theoretical calculations (solid lines). The magnetization measurements in large d.c. fields of 1.5 and 3.0 T (typical fields for clinical magnetic resonance imaging systems) show a substantial change near body temperature (37 °C).

rapid fluctuations[35–38]. Here, the additional line broadening comes from the static (in time) perturbation of the external field by the inhomogeneous dipole field of magnetic particles.

Before discussing the details of our model, we evaluated whether such a static model is appropriate. Similar to Brown's[39] and Yablonskiy's[40] papers, we assumed that protons are effectively static. To validate this assumption, we analysed motional narrowing limits for the ferromagnetic particles that were used[41]. We employed the published values of a self-diffusion coefficient in normal water at different temperatures[42,43] because diffusion in 1% concentration agar gels is only about 15% smaller when compared with water[44]. We calculated that in the temperature range between 5 and 35 °C, $\Delta\omega \cdot \tau_D \geq 1$, where $\Delta\omega = \gamma\Delta H$, $\tau_D = \frac{d^2}{4D}$, $\Delta H$ is the magnetic field-linewidth, $\gamma$ is gyromagnetic ratio for a proton, $D$ is a self-diffusion coefficient, and $d$ is the average Gd particle size used in experiments. The magnetic field inhomogeneity, measured by $\Delta H$, comes from the temperature-dependent fields due to the magnetic particles (calculated below). We concluded, therefore, that we are in the static dephasing regime, where the diffusion of water molecules may be neglected.

Based on the concentration of the magnetic particles, we can estimate an effective volume/particle. This three-dimensional volume is then broken into multiple cells and we calculate the magnetic field for each cell. One then makes a histogram showing the number of cells with a given effective field $H_{eff}$. The linewidth can then be directly found from the histogram as the full-width at half maximum.

The results of the calculation are shown as solid lines in Fig. 3a. The theoretical results are in good quantitative agreement with the experimental results in terms of both the temperature and concentration dependences. The calculations are based on Gd particles (diameter 5 μm), but where the magnetization of the particles has been reduced by a factor of 5.1 to match the magnetization experiments as discussed earlier. In practice, small Gd particles and films generally have reduced magnetizations due to voids, impurities, oxidation and surface effects[45–48]. Figure 3b shows a comparison between the numerical values we obtained and the analytical results calculated using Brown's formula. The results are almost identical. A more detailed description of the calculations is included in the Methods (MRI temperature measurements section).

We used the gradient echo method for MRI imaging because it is very sensitive for local magnetic field inhomogeneity[49]. Images of the phantom consisting of pure Ringer's solution-agar gel (top row) and Ringer's solution-agar gel with 2.75 mMl$^{-1}$ Gd concentration (bottom row) are shown in Fig. 4. Note that images of agar gel with Gd show a strong temperature-dependent increase in brightness.

Figure 5 presents the temperature-dependent relative MR intensity (the ratio of image intensity of pure agar gel to the image intensity of gel doped with Gd particles) for different Gd concentrations. It is clear from this data that several different concentrations of Gd allow for temperature-dependent measurements.

We can estimate the accuracy of the measured temperature. The ratios of MR image intensities from 10.8 °C through 39.1 °C (Fig. 5) for all concentrations of gadolinium were statistically analysed using a regression of the means. From the regression's 95% confidence bands, we estimated the accuracy of the temperature determination in the phantom. Figure 6 shows a summary of the temperature measurement accuracy in the narrowest region of confidence bands (24 °C) and in the region of human body temperature (37 °C). The best temperature resolution of ± 0.6 and ± 1.0 °C was achieved for a concentration of 0.69 mMl$^{-1}$ at 24 °C, and at 37 °C, respectively.

Using NMR, we measured $T_1$ and $T_2$ nuclear relaxation times in pure Ringer's solution-agar gel and in Ringer's solution-agar gel with different concentrations of Gd particles. The relaxation time results for the highest Gd concentration (2.75 mMl$^{-1}$) are given in Table 1. We observed only small changes in the $T_2$ behaviour for samples with and without Gd. The change in $T_1$ was essentially independent of the presence of Gd. The lack of sensitivity in $T_1$ is consistent with the idea that the Gd provides a static perturbation because $T_1$ is generally affected by fast dynamic changes in the environment of the protons. In contrast, the change in NMR linewidth, originating from the thermal decrease of the Gd particle magnetization, showed a 269%

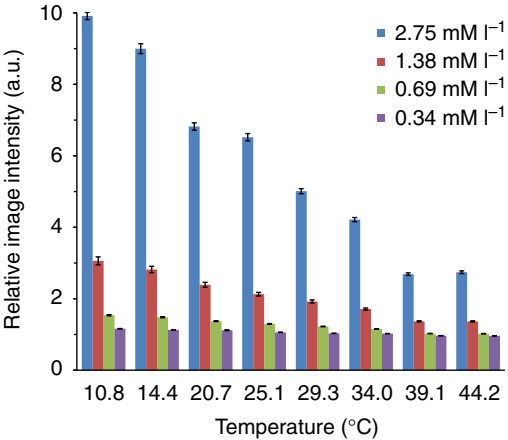

**Figure 5 | Temperature dependence of magnetic resonance image intensities.** The ratio of the mean $T_2^*$ weighted image intensities for samples of pure agar to samples of agar with Gd particles as a function of temperature for various Gd concentrations. Means and standard deviations were obtained after averaging the intensities in the region of interest, consisting of 195 voxels within the image center.

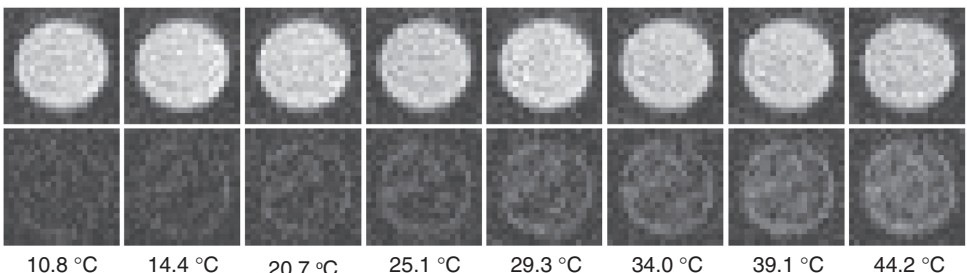

**Figure 4 | Magnetic resonance images of the phantom at different temperatures.** Top row: pure 1% Ringer's solution-agar gel. Bottom row: 1% Ringer's solution-agar gel doped with the highest content of Gd particles (2.75 mMl$^{-1}$). Circular objects on axial images are 10 mm across. Note that the image of the agar gel with the Gd particles (bottom) shows a dramatic brightening as the temperature is increased.

decrease as the temperature was increased from 10 to 40 °C as seen in Fig. 3a. From these observations, and because $T_2^* = \frac{1}{\pi \Delta f}$, where $\Delta f$ is the observed linewidth[50], we concluded that $T_2^*$ would be a much more sensitive parameter for detecting temperature changes than $T_1$ or $T_2$ using MRI imaging[29].

The gradient echo images of phantoms at 1.5 T with various concentrations of Gd particles in agar gel show significant changes in image intensity as a function of temperature. Using a Gd concentration of only 0.69 mM l$^{-1}$ allows for temperature determination with an accuracy under one degree Celsius in the temperature range 11–37 °C. In comparable studies using TmDOTMA, James et al.[26] obtained an accuracy of 0.3 °C in temperatures between 30 and 44 °C. However, these measurements were carried out at 9.4 T on an agar gel phantom with a 16 mM l$^{-1}$ concentration of the paramagnetic thulium complex. We point out that our method has significant advantages. First, a substantially smaller concentration of contrast material is needed; the ferromagnetic particles require about a factor of 20 less material. Second, the magnetic fields used in our method are significantly lower; in fact, they are the fields used in current clinical scanners.

Our thermal resolution can be improved in a number of different ways. Different compositions (alloys and heterogeneous structures) and sizes of magnetic particles will change the temperature-dependent MR image contrast. For example, our theoretical calculations indicate that adding 5% of Co to the Gd can increase the Curie temperature by 30–40 °C, resulting in a more rapid change of $M(T)$ near body temperature. Based on these calculations, we estimate a potential temperature resolution of 0.1–0.2 °C.

Other materials may be designed to work in a variety of temperature ranges. For example, Permalloy (Fe$_{0.2}$Ni$_{0.8}$) normally has a Curie temperature of 576 °C. However, Cu doping (48.5%) can reduce $T_C$ to 43 °C (refs 51,52). We have also made films of

Fe$_{0.2}$Ni$_{0.8}$ which were 50% doped with Cu. The Curie temperature in this case was ∼55 °C, indicating that this may be a less toxic and more biocompatible alternative for Gd. Another possibility is to use an Fe/Cr alloy[53].

Future biological studies are needed to address a variety of other issues including particle size, toxicity[54–56], stability and delivery methods. Our theoretical calculations show that the expected NMR linewidth broadening can work equally well for smaller (sizes below a micron), but more numerous ferromagnetic particles. In terms of toxicity, we note that Gd contrast agents have been approved by the Federal Drug Administration (FDA) for use in humans provided that the Gd is adequately sequestered. This can be accomplished by using biocompatible coatings such as gold[57], silicon dioxide[58] or dextran[59].

The method demonstrated here shows significant promise. However, to know the absolute temperature, one also needs to know the concentration of the magnetic particles. This is true in some other current methods as well[16]. This is clearly possible on the surface of implants or in materials containing a prescribed concentration, but would be more difficult in vivo. However, with an unknown concentration, it is possible to measure temperature differences, for example differences or changes introduced by local heating during hyperthermia procedures.

The results presented above were obtained with Gd particles that were not protected from deterioration. As stated in the Methods section, we monitored the linewidth of Gd in agar gel over time and found an 8 Hz per month reduction in linewidth (about 2.1% of initial value per month) indicating a progressive diminishing of the particle's metallic core and consequently lowering its magnetization. Because the SQUID, NMR and MRI experiments were conducted within the first three weeks, the deterioration of Gd did not influence our results. However, a gradual deterioration of metallic Gd particles may cause serious problems for long-term particle storage. To prevent Gd particles from degradation, we developed a new technology to produce particles covered with a protective layer of 5 nm chromium by photolithography and magnetron sputtering lift-off process. Samples of 1% agar Ringer's solution gel with such particles were tested for NMR linewidth broadening immediately after fabrication and after 7 months of storage at 4 °C. We found only a 6% decrease of linewidth, a range within experimental error of our NMR setup. We conclude that coating helps to preserve the magnetic properties of metallic particles which are fundamental to this concept. Another consideration might be to coat the particles with other materials such as gold, silica or dextran which will also address toxicity issues.

The successful development of this method will allow for a noninvasive temperature measurement inside bodies and other materials using low concentrations of magnetic particles. This method may be used for detecting and monitoring inflammation and tumorous activity. In addition, different shades of grey in the MRI images could be calibrated to obtain a map of temperature or to report the achievement of a certain temperature threshold in a specific tissue during interventional procedure. In addition, patients with metallic implants cannot normally undergo MRI

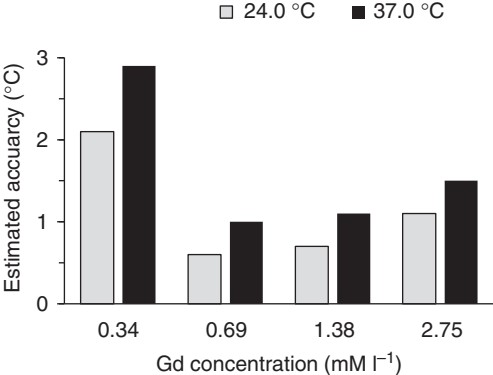

**Figure 6 | Accuracy in the temperature determination for different Gd concentrations.** Accuracy was determined from a linear regression of intensity changes of MR images for the different Gd particle concentrations. Note that the highest sensitivity is not achieved for the highest Gd concentration.

**Table 1 | Nuclear relaxation times for phantoms with and without Gd.**

| $T$ (°C) | $T_1$ (ms) | | $T_2$ (ms) | |
|---|---|---|---|---|
| | Pure agar | Agar with Gd | Pure agar | Agar with Gd |
| 10.0 | 1,104 ± 35 | 962 ± 23 | 181.0 ± 3.6 | 104.9 ± 2.2 |
| 40.0 | 1,919 ± 31 | 1,346 ± 42 | 170.3 ± 2.5 | 118.7 ± 1.2 |

Values of $T_1$ and $T_2$ in pure agar gel and agar gel with Gd particles (concentration 2.75 mM l$^{-1}$) at 10 and 40 °C. Note that the relative change in $T_1$ with temperature is not sensitive to the presence of Gd. The changes in $T_2$ as a function of temperature are very small in all cases.

scanning because one cannot assess the temperature increases of the tissue near the implants during imaging. Coating medical metallic implants with magnetic elements during their fabrication may allow direct identification of the temperature near the implant site, eliminate issues of particle delivery or side effects, and will make MRI procedures possible for a large group of people[60,61].

## Methods

**NMR and MRI sample preparation.** For the [1]H NMR and MRI measurements, the Gd particles were suspended in a 1% Ringer's solution-agar gel. This created an isotonic solution similar to the bodily fluids of an animal and prevented particles from sedimentation. A mixture containing 40 cc of 1% Ringer's solution-agar gel and 17.2 mg of Gd particles was prepared (2.75 mM l$^{-1}$ concentration). Portions of this sample were subsequently diluted with additional 1% Ringer's solution-agar gel to obtain mixtures containing 1.38, 0.69 and 0.34 mM l$^{-1}$ of Gd concentration. The mixtures were kept in a liquid state at 90 °C in a water bath and constantly

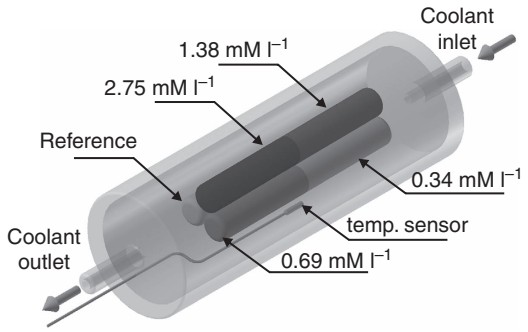

**Figure 7 | Diagram of the magnetic resonance imaging temperature setup.** The setup contains Gd-Ringer's solution-agar gel phantom with four different concentrations of Gd particles and pure Ringer's solution-agar gel used as reference. The temperature of the phantom was controlled by the flow of perfluorocarbon coolant and was monitored by a miniature fibre optic sensor.

stirred before transferring to standard 5 mm glass tubes for the NMR studies (of 0.12 cc volume) and to Nalgene plastic vials (4.5 cc volume and 10 mm diameter) to make an MRI phantom. Mixtures were rapidly cooled in ice water to preserve an even distribution of gadolinium particles in the gel.

**Magnetization measurements.** The magnetization of the Gd particles was measured in the range of 0–60 °C at different magnetic fields using a super-conducting quantum interference device magnetometer to determine the temperature dependence of the magnetization and the Curie temperature. Additional SQUID measurements of a bulk, 0.8 mg sphere of Gd were conducted for the assessment of the magnetization reduction in Gd particles used for NMR and MRI due to the presence of voids, impurities, oxidation and other surface processes. We obtained a 5.1 ratio of bulk magnetization to particles magnetization. This single parameter was later used in the theoretical calculations of magnetization and NMR linewidth.

As noted the measured magnetization of the particles is substantially lower than that expected for bulk single crystal Gd. In fact, the magnetization of Gd can be significantly reduced by a number of different origins. As noted in Scheunert *et al.*[46] a few per cent of oxygen in the Gd can reduce the magnetization by 20%. There are reports, spanning many years, showing the magnetization of thin films of Gd often is significantly smaller than seen in single crystal samples[46,47]. Gd films deposited at room temperature typically have smaller grain sizes, a larger proportion of a paramagnetic fcc Gd phase and residual strain in the sputtered films, all resulting in a lowered magnetization[46]. Thin films with a magnetic response akin to single crystals were found to have strong intergranular interactions[46]. Obviously weaker intergranular interactions lead to lower magnetizations.

To further explore the nature of the discrepancy between the bulk magnetization and that found in our sample, we performed energy dispersive X-ray spectroscopy on gadolinium particles and found the presence of oxygen with a value of about 10%, indicating a substantial reduction in magnetization was to be expected[46]. In addition to the presence of oxygen we have shown that our samples are polydisperse in size and shape suggesting the presence of multiple grain boundaries. We find that these factors, in agreement with the previous references[46–48], could account for a significant reduction in magnetization.

**NMR measurements.** The proton nuclear $T_1$ and $T_2$ relaxation times, and proton linewidth in Ringer's solution-agar gel with and without Gd particles were measured with a low-field (364 mT/15.5 MHz) pulsed spectrometer. The results were used as a guidance in the selection of useful Gd concentrations for the MRI phantoms. $T_1$ was obtained using the inversion recovery method and $T_2$ using the Carr-Purcell-Meiboom-Gill (CPMG) sequence. The linewidth (FWHM) was

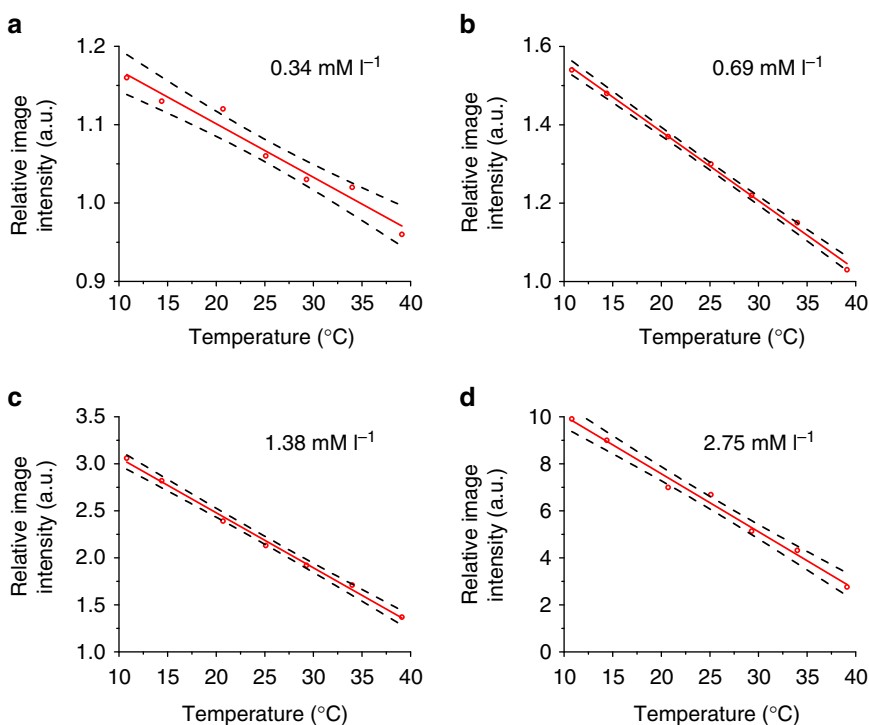

**Figure 8 | Linear regression for MRI intensity versus temperature.** (**a**–**d**) Experimental relative image intensities (red circles) and corresponding regression plots (red solid line) versus temperature for various Gd concentrations. The corresponding 95% confidence bands (black dashed lines) are used for the calculation of the temperature accuracy.

determined from the free induction decay (FID) Fourier Transform. During the temperature-dependent NMR measurements, the samples were cooled and $^1$H NMR spectra were taken after the temperature was stabilized from 5 to 50 °C every 5 °C with accuracy of ± 0.15 °C.

**MR imaging.** The MRI temperature-dependent images of the phantom containing pure 1% Ringer's solution-agar gel and four different concentrations of Gd particles in 1% Ringer's solution-agar gel were obtained using a preclinical scanner with a dc magnetic field of 1.5 T and a 30 cm bore magnet equipped with a temperature control system. A schematic diagram of the temperature setup is shown in Fig. 7.

The phantom consists of three Nalgene plastic vials (10 mm inner diameter and 80 mm long) placed inside a polycarbonate cell. One vial contains 1% Ringer's solution-agar gel and serves as a reference. Two other vials contain four different concentrations of Gd particles suspended in 1% Ringer's solution-agar gel (two Gd concentrations in one vial). The continuous flow of proton-less perfluorocarbon coolant through the cell forced by a standard circulating bath stabilizes the phantom temperature without contaminating $^1$H images with additional signals. For imaging, a multi-slice gradient echo sequence (GRE) was used with the following imaging parameters: field of view = $3 \times 3$ cm, slice thickness = 3 mm, matrix = $64 \times 64$, echo time TE = 2.5 ms. Twelve axial slices were taken. A long repetition time of TR = 5.0 s was used to avoid signal loss due to a relatively long $T_1$ relaxation time in the phantom. The phantom temperature was monitored by a signal conditioner connected by fibre optic guide to a miniature, high-precision birefringent sensor placed in the space near the Nalgene vials. The sensor was calibrated to ± 0.15 °C accuracy within the measurement range.

**Image intensity analysis.** The mean intensity of the MRI images and the corresponding standard deviation were calculated across the entire axial slice using a Matlab platform programme. The accuracy of the temperature measurements was determined as the range within the 95% confidence interval lines obtained from linear regression analysis as shown in Fig. 8.

**Numerical calculation of NMR linewidth.** The calculation was initially done with a large cube containing a single magnetic particle at the centre. The cube was partitioned into cells of size 1 μm on a side. The number of cells in the large cube was varied so that the side of the large cube was equal to the average spacing between magnetic particles. Thus different concentrations resulted in different volumes for the large cube, where the smallest value was $(107 \, \mu m)^3$ for the 2.75 mM l$^{-1}$ concentration and the largest was $(214 \, \mu m)^3$ for the 0.34 mM l$^{-1}$ concentration. A magnetic dipole was located at the centre of the cube and the magnetic field values were calculated for all cells in the cube as a function of position from this dipole, from which the linewidth is calculated as discussed in the text.

A number of additional simulations were done where the magnetic dipole was placed randomly within the cube or multiple magnetic dipoles were randomly placed within a larger volume, keeping the concentration constant. The particles are relatively far apart, and as a result the dipole field of one particle acting on a neighbouring particle is quite small, about $10^4$ times smaller than the applied field in the NMR experiments. Thus dipole–dipole interactions can safely be neglected. Consistent results were obtained for linewidth as a function of concentration for all these numerical experiments.

**Data availability.** The authors declare that all relevant data are available on request.

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

## Acknowledgements

This work was supported by the State of Colorado Bioscience Discovery Evaluation Grant (14BGF- 22) and the UCCS BioFrontiers Center. We thank Dr Y. Garbovskiy, J. Baptist, J. Nobles and K. Smiley from UCCS for their contribution to the project, Drs T.K. Yasar and T.J. Royston from University of Illinois at Chicago for making the Matlab programme available for image intensity calculations, and Drs P. Kabos and S. Russek from National Institute of Standards and Technology in Boulder, CO, for their valuable comments on the manuscript. This paper is a contribution of NIST and is not subject to copyright in the United States.

## Author contributions

J.H.H. designed experiments, performed SQUID and NMR measurements, and conducted data analysis, Z.C. developed the experimental concepts and provided particles, K.F.S. performed acquisition of temperature MRI images and R.C. and N.A. developed the theory and performed theoretical calculations. All authors have discussed results and contributed to writing of the paper.

## Additional information

**Competing financial interests:** The authors declare no competing financial interests.

