## [Peer Review File · Nature Communications]

Reviewer #1 (Remarks to the Author):

The authors have modified the manuscript and have followed many of my remarks.

I still have two concerns.

The first is about the relaxation theory of Brown that the authors have now introduced in their interpretation:

the authors state that in the relaxation equation $\Delta(\omega) = 2\pi \cdot \text{linewidth}$. It is wrong. $\Delta(\omega)$ is related to the magnetization of the particles and should be calculated as described in the literature. you cannot use the experimental nmr result to validate the nmr relaxation theory you want to use to interpret your NMR results...

My other concern is the lack of explanation for the 5 fold decrease of the magnetization. The reference given by the authors concerns data at very low temperatures (4.2K), and for very small particles compared to the ones used in this study.

Minor comment:

in my previous remarks, I asked the authors to replace T2* maps by T2 weighted images. It has to be T2* weighted images. Sorry for the mistake.

Reviewer #2 (Remarks to the Author):

The revised version of the manuscript by J. H. Hankiewicz et al. "Ferromagnetic particles as MRI temperature sensors" has clarified most issues raised in my previous report and in the reports of other referee. In my opinion, it can be published in Nature Communications, after some minor revisions.

1) Most importantly, it would be very helpful if the authors provided details of the numerical calculations leading to the results shown in Fig. 3. How many cells were used in the simulations? How many particles per cell? How were the particles distributed over the simulated volume? Did the authors average over different distributions of the particles over the volume? Were the long-range dipole-dipole interactions taken into account? How did the authors check that their simulations produced sufficient statistics to adequately calculate the distribution of the magnetic fields in the sample? How exactly was the correcting factor of 5.1 applied in the simulations? All this information can be included in the Supplementary information to meet the limitation on the length of the paper, but it is important to have these details for reproducibility of the results.

2) After taking into account the correction factor of 5.1, the theoretically calculated curves in Fig. 3a still noticeably differ from the experimental results. Would agreement be better if, instead of the correction factor 5.1, the author used the experimentally measured magnetization (shown in Fig. 2), keeping the rest of the calculation method unchanged? How would the theory-experiment comparison look in that case? I think such information would be very useful for future efforts to deeper understand the mismatch between theory and experiment. Again, this part could be communicated in the Supplementary information to keep the paper length intact.

3) I really like the comparison shown in Fig. 3b: it answers several questions about comparison between the work of Brown and the simulations made in the present manuscript.

Reviewer #1 (Remarks to the Author):

The authors have modified the manuscript and have followed many of my remarks. I still have two concerns.

Comment 1. The first is about the relaxation theory of Brown that the authors have now introduced in their interpretation: the authors state that in the relaxation equation $\Delta\omega = 2\pi \cdot \text{linewidth}$. It is wrong. $\Delta\omega$ is related to the magnetization of the particles and should be calculated as described in the literature. you cannot use the experimental nmr result to validate the nmr relaxation theory you want to use to interpret your NMR results.

Response: We agree the justification for the static dephasing assumption should be based off of theoretical calculations. We now use $\Delta\omega = \gamma\Delta H$ where, as the referee suggested, the ΔH comes from the temperature-dependent fields due to the magnetic particle. This still gives a $\tau_d\Delta\omega \geq 1$ for the temperature range of interest. The relevant section in the paper has been changed to show that this is based on calculation rather than experimental measurement.

We have changed the following in the paper on page 7:

“We calculated that in the temperature range between 5 °C and 35 °C, $\Delta\omega \cdot \tau_D \geq 1$, where $\Delta\omega = \gamma\Delta H$, $\tau_D = \frac{d^2}{4D}$, ΔH is the magnetic field-linewidth, γ is gyromagnetic ratio for a proton, D is a self-diffusion coefficient, and d is the average Gd particle size used in experiments. The magnetic field inhomogeneity, measured by ΔH , comes from the temperature-dependent fields due to the magnetic particles (calculated below). We concluded, therefore, that we are in the static dephasing regime, where the diffusion of water molecules may be neglected.”

Comment 2. My other concern is the lack of explanation for the 5 fold decrease of the magnetization. The reference given by the authors concerns data at very low temperatures (4.2K), and for very small particles compared to the ones used in this study.

Response: We agree that the reduction of Gd magnetization is surprising. As the referee noted, the reference previously given does concern data at 4.2 K, however presumably, the magnetization reduction should continue or become larger at higher temperatures.

We have included three additional references showing significant magnetization reductions in Gd films:

[46] Scheunert, G., Hendren, W. R., Ward, C. & Bowman, R. M. Magnetization of 2.6 T in Gadolinium thin films. *Appl. Phys. Lett.* **101**, 142407 (2012).

[47] Yamada, Y., Okada, M., Jin, P., Tazawa, M. & Yoshimura, K. The Curie Temperature Dependence on preparation conditions for Gd thin films. *Thin Solid Films* **459**, 191-194 (2004).

[48] Romera, M., Munoz, M., Maicas, M., Michalik, J. M., de Teresa, J. M., Magen, C., & Prieto, J. L. Enhanced exchange and reduced magnetization of Gd in an Fe/Gd/Fe trilayer. *Phys. Rev. B* **84**, 094456 (2011).

To further address the referee's concerns, we have added the material below in the **Methods. Magnetization measurements** section of the paper on page 13:

“As noted the measured magnetization of the particles is substantially lower than that expected for bulk single crystal Gd. In fact, the magnetization of Gd can be significantly reduced by a number of different origins. As noted in Ref [46] a few percent of oxygen in the Gd can reduce the magnetization by 20%. There are reports, spanning many years, showing the magnetization of thin films of Gd often is significantly smaller than seen in single crystal samples^{46, 47}. Gd films deposited at room temperature typically have smaller grain sizes, a larger proportion of a paramagnetic fcc Gd phase and residual strain in the sputtered films, all resulting in a lowered magnetization⁴⁶. Thin films with a magnetic response akin to single crystals were found to have strong intergranular interactions⁴⁶. Obviously weaker intergranular interactions

lead to lower magnetizations.

To further explore the nature of the discrepancy between the bulk magnetization and that found in our sample we performed energy dispersive x-ray spectroscopy (EDX) on gadolinium particles and found the presence of oxygen with a value of about 10 %, indicating a substantial reduction in magnetization was to be expected⁴⁶. In addition to the presence of oxygen we have shown that our samples are polydisperse in size and shape suggesting the presence of multiple grain boundaries. We find that these factors, in agreement with the previous references^{46, 47, 48}, could account for a significant reduction in magnetization.”

Comment 3. Minor comment: in my previous remarks, I asked the authors to replace T2* maps by T2 weighted images. It has to be T2* weighted images. Sorry for the mistake.

Response: Yes. We agree that using the term “T2* weighted” instead of “T2 weighted images” will describe the imaging method more precisely. We made necessary changes in the text and figure captions accordingly.

Reviewer #2 (Remarks to the Author):

The revised version of the manuscript by J. H. Hankiewicz et al. "Ferromagnetic particles as MRI temperature sensors" has clarified most issues raised in my previous report and in the reports of other referee. In my opinion, it can be published in Nature Communications, after some minor revisions.

Comment 1. Most importantly, it would be very helpful if the authors provided details of the numerical calculations leading to the results shown in Fig. 3. How many cells were used in the simulations? How many particles per cell? How were the particles distributed over the simulated volume? Did the authors average over different distributions of the particles over the volume? Were the long-range dipole-dipole interactions taken into account? How did the authors check that their simulations produced sufficient statistics to adequately calculate the distribution of the magnetic fields in the sample? How exactly was the correcting factor of 5.1 applied in the simulations? All this information can be included in the Supplementary information to

meet the limitation on the length of the paper, but it is important to have these details for reproducibility of the results.

Response: We agree that including the simulation details in the text will definitely improve the manuscript. We have added the following to the **Methods** section on page 15:

“Numerical calculation of NMR linewidth. The calculation was initially done with a large cube containing a single magnetic particle at the center. The cube was partitioned into cells of size 1 micron on a side. The number of cells in the large cube was varied so that the side of the large cube was equal to the average spacing between magnetic particles. Thus different concentrations resulted in different volumes for the large cube, where the smallest value was (107 microns)³ for the 2.75 mM/L concentration and the largest was (214 microns)³ for the 0.34 mM/L concentration. A magnetic dipole was located at the center of the cube and the magnetic field values were calculated for all cells in the cube as a function of position from this dipole, from which the linewidth is calculated as discussed in the text.

A number of additional simulations were done where the magnetic dipole was placed randomly within the cube or multiple magnetic dipoles were randomly placed within a larger volume, keeping the concentration constant. The particles are relatively far apart, and as a result the dipole field of one particle acting on a neighboring particle is quite small, about 10^4 times smaller than the applied field in the NMR experiments. Thus dipole-dipole interactions can safely be neglected. Consistent results were obtained for linewidth as a function of concentration for all these numerical experiments.”

Comment 2. After taking into account the correction factor of 5.1, the theoretically calculated curves in Fig. 3a still noticeably differ from the experimental results. Would agreement be better if, instead of the correction factor 5.1, the author used the experimentally measured magnetization (shown in Fig. 2), keeping the rest of the calculation method unchanged? How would the theory-experiment comparison look in that case? I think such information would be very useful for future efforts to deeper understand the mismatch between theory and experiment. Again, this part could be communicated in the Supplementary information to keep the paper length intact.

Response: Indeed, we have performed the calculations using the experimental magnetization at each temperature in the linewidth calculations and some of the theoretical and experimental curves are closer and look more similar in character (see Fig. 1 below), which shows the best case. The results for the other concentrations were not as good, however. It seems simpler and more appropriate to use the theoretical values throughout (with the appropriate reduction factor from the bulk) as this allows a full comparison of all the experimental data with theoretical results based on a single parameter.

Figure 1: A comparison between experimental values for linewidth at a concentration of 2.75 mM/L with theory. The theory curves were calculated using the temperature dependent magnetization from the experimental measurements presented in the manuscript.

Comment 3. I really like the comparison shown in Fig. 3b: it answers several questions about comparison between the work of Brown and the simulations made in the present manuscript.

Response: We thank the reviewer for this comment. We agree that simultaneous plotting of data from Brown's work and our theoretical data allows for a quick comparison of both methods.

We again thank the referees and editor for their helpful comments. We made all the changes suggested by both the referees and the editor. We hope this manuscript can now be accepted in Nature Communications.

Authors